# Factors Driving Autumn *Quercus* Flowering in a Thermo-Mediterranean Area

**Herminia García-Mozo** , **Rocío López-Orozco** , **Jose Oteros and Carmen Galán** *

Department of Botany, Ecology and Plant Physiology, Agrifood Campus of International Excellence CeiA3, University of Cordoba, Rabanales Campus, E-14071 Córdoba, Spain
* Correspondence: bv1gasoc@uco.es

**Abstract:** The flowering period of plants is a critical time since it determines their reproductive success. Flowering is controlled by different factors including genetic regulation and environmental conditions. In the Mediterranean area, favourable conditions usually occur in spring, when most plant species flower including those of the Mediterranean *Quercus* genus. This paper reveals and analyses an unusual and lesser-known phenomenon occurring in the two main Mediterranean agroforestry ecosystems of South Europe, the Mediterranean forest and "dehesa", that is, a second flowering occurring in autumn for the species *Quercus ilex* subsp. *ballota* (holm oak). The continuous pollen monitoring of the atmosphere in the city of Cordoba (southern Spain) for 25 years, together with field phenological observations in the area, has indicated that, apart from the main pollination period in spring, secondary flowerings also occasionally occur in this area, specifically in autumn. The present work examines these uncommon pollination events detected in the autumns of certain years with the aim of determining the main environmental factors that influence and control them. During the 25-year study period, there were 7 years in which a secondary *Quercus* flowering was detected in the area from the second half of October until the end of November. The univariate statistical analysis of the influence of environmental variables determined that the meteorological conditions in September were the most influential. Low mean temperatures, together with record rainfall in that month, led to autumn flowering events. The phenological characteristics of the spring pollen season were also influential. In the years with a shorter spring, the *Quercus* pollen season tended to present autumn flowerings. A multivariate adaptive regression splines (MARS) model was built to explain the effects of the different variables on the occurrence of autumn pollination. The results indicated that the combined effect of three predicting variables, September rainfall, the length of the spring pollen season, and the end of the spring pollen season, explained 92% of the variance. The validation showed a strong relationship between the expected and the observed autumn pollen concentrations. Therefore, the present analysis of a long-term pollen database revealed that the main causes of this unusual second flowering in autumn were strongly related to climate change, i.e., strong dry summers and warm autumns. In addition, the results showed that the phenomenon was more frequent in the years with low pollination during spring due to different meteorological events potentiated by climate change, such as dryness or heavy rain episodes, as a way of ensuring acorn crops. The results explain how this unusual and lesser-known phenomenon in agroforestry dynamics is related to the adaptation to climate change and the main factors that are driving it, as well as the potential consequences for these important and endangered Mediterranean ecosystems.

**Keywords:** pollen; *Quercus*; holm oak; autumn flowering; secondary pollination



## 1. Introduction

The timing of life-cycle events throughout the year is a decisive feature allowing plants' adaptation to seasonally changing environments [1–3]. The flowering period of plants can be considered the most critical time since it determines their reproductive success. Flowering is controlled by several endogenous and exogenous factors, including genetic

regulation and environmental conditions [4–6]. These involve coordinating flowering with the appropriate season, depending on the different pollination pathways of the plants. In the case of anemophilous plants, favourable conditions are commonly associated with low humidity and warm temperatures. In the Mediterranean area, these conditions usually occur in spring, when most plant species flower, including those of the Mediterranean *Quercus* genus. These are anemophilous species that produce and release high quantities of pollen into the atmosphere during the spring from March to June [7,8]. *Quercus* species are frequently found in natural Mediterranean areas, to the extent that some of them are the dominant tree species of the main Mediterranean ecosystems, the Mediterranean forest and the "dehesa". The "dehesa" is an anthropic-influenced landscape of meadows, where trees grow in a low density and where the ground is covered with a blanket of grasses and other herbaceous species [9]. The *Quercus* fruits (acorns) and the grazing pastures of the "dehesas" make them an important sustainable resource for livestock farming in southern Europe, especially in Spain and Portugal. The main *Quercus* species in these areas are the holm oak (*Quercus ilex* subsp *ballota* (Desf. Samp.)), the cork oak (*Quercus suber* L.), the kermes oak (*Quercus coccifera* L.), and the Portuguese oak (*Quercus faginea* L).

Although some authors have indicated the occurrence of autumn flowering in Quercus trees including in aerobiological studies, for many years, the causes of this phenomenon in the Mediterranean area have remained unknown [8]. In fact, most of the related literature indicates that spring is the season in which the *Quercus* genus (including Mediterranean species) flowers, and autumn is the season in which acorns mature but they do not give importance to this secondary flowering [7,10–12]. Regarding other species, it has been observed that plants living in Mediterranean-type climates, whose main favourable period for plant photosynthetic activity is spring, may flower in both spring and autumn under certain conditions [13,14]. Nevertheless, background information regarding this phenomenon of secondary flowering is scarce. In the case of Mediterranean evergreen *Quercus* species, some studies indicate the high resilience of their photosynthetic apparatus to summer droughts, along with a good recovery in the following autumn rains [15]. Secondary flowering in autumn has been mostly mentioned for holm oak and occasionally for cork oak, which may have a two-year fruit maturation cycle [16].

Certain authors have attempted to explain this in relation to unusual vegetative growing that some individuals may experience under certain climatic conditions in autumn, but the specific factors controlling this phenomenon have not been revealed until now [16,17].

In the present work, the continuous aerobiological pollen monitoring of the atmosphere in the city of Cordoba (southern Spain) for 25 years, which is surrounded by *Quercus* Mediterranean forests, together with field phenological observations in the area, has indicated that, apart from the main pollination period in spring, secondary flowerings also occasionally occur in this area, specifically in autumn. The present work examines these uncommon pollination events detected in the autumns of certain years with the aim of determining the main environmental factors that influence and control them.

The principal objective of the present research was to study this natural but infrequent phenomenon of *Quercus* autumn flowering by reviewing an extensive atmospheric pollen database (over a 25-year period) owing to the anemophilous character of this genus. Moreover, three specific objectives were proposed to attain a full understanding of the phenomenon:

- 　To detect the most remarkable *Quercus* autumn events during the study period.
- 　To determine the main environmental factors influencing autumn flowering.
- 　To propose a forecasting model that is capable of predicting the occurrence of this secondary flowering phenomenon based on an analysis of multivariate interactions.

## 2. Materials and Methods

### 2.1. Study Area

The study was carried out by analysing 25 years of daily airborne pollen data from 1995 to 2019 from monitoring carried out in the city of Cordoba (37°53′ N, 4°47′ W),

located in the region of Andalusia, Spain, 111 m.a.s.l. (Figure 1). Cordoba is a small city (325,000 inhabitants) situated in southern Spain in the valley of the Guadalquivir River. It is surrounded by the Sierra Morena Mountain chain to the north (200–600 m.a.s.l.), where the landscape is characterised by Mediterranean oak forests and 'dehesa', a human-managed pasture landscape in which the dominant woody species are perennial oaks. The most abundant *Quercus* species in the Sierra Morena are *Quercus ilex* subsp. *ballota* (Desf.) Samp. (holm oak); followed by *Quercus suber* L. (cork oak) in the wetter areas with acid soils; *Quercus coccifera* L. (kermes oak), which appears in the sunniest areas of *Quercus* mixed ecosystems with poor soils; and *Quercus faginea* Lam. (Portuguese oak), a marcescent species that grows in highly specific humid areas with basic soils. The distribution of these species is shown in Figure 1. All of them are anemophilous species that are pollinated by the wind and have high pollen production, and all of them belong to the *Quercus* pollen type [18]. Nevertheless, the most important contributor to the atmospheric *Quercus* pollen concentrations is the holm oak owing to its wide distribution compared with the other *Quercus* species in the study area.

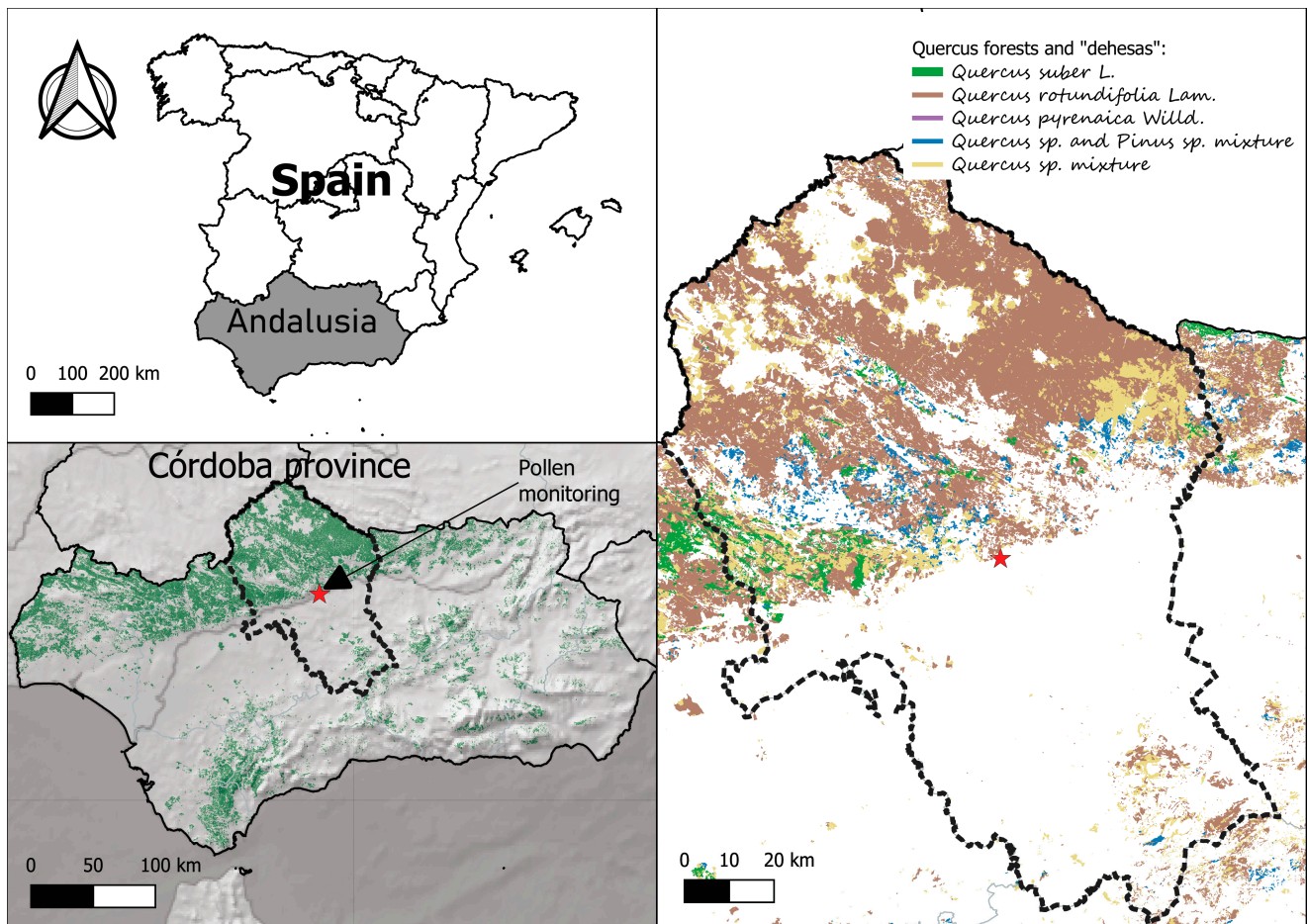

**Figure 1.** Study area and location of the pollen monitoring station in Cordoba city (37°53′00″ N 4°46′00″ W) are indicated in relation to the *Quercus* landscape distribution in the Andalusia Region and Cordoba province (right picture). Source: Plan Forestal Andaluz, 2018 [19].

### 2.2. Scientific Data

2.2.1. Aerobiological and Phenological Data

Atmospheric pollens that were monitored for 25 years from 1995 to 2019 were analysed using a volumetric Hirst-type spore trap [20] placed 20 m above ground level in the city of Cordoba, which is surrounded by *Quercus* Mediterranean forests (Figure 1). The method

and location chosen are those recommended by the European Aerobiology Society (EAS), as is the analysis of pollen grains under light microscopy [21,22].

The study of *Quercus* pollen emissions was based on the analysis of different aerobiological parameters following the EAS recommendations. The main pollen season (MPS) is the period in which significant concentrations of pollen occur, starting with the first of 5 consecutive days, with $\geq 10$ pollen grains/m$^3$, and ending with the last of 5 consecutive days, with $\leq 10$ pollen grains/m$^3$ [7]. With respect to the MPS, certain features were considered, such as its length in number of days, the peak value (peak) as the maximum daily pollen concentration (pollen grains/m$^3$), and the peak date as the day of the year on which the peak value was recorded. The length of the pre-peak season as the number of days from when the pollen season started to the peak date, and the length of the post-peak season as the number of days from the peak date to the end of the pollen season, were also analysed. These parameters were additionally studied during the secondary autumn pollen season.

With respect to the characteristics of the pollen emission intensity, the features analysed were the annual pollen integral (APIn) as the sum of the daily pollen concentrations for the whole year (pollen grains * day/m$^3$), the main pollen season integral (MPSIn), the pre-peak season integral (PrPSIn), and the post-peak season integral (PsPSIn) as the sum of the daily pollen concentrations (pollen grains * day/m$^3$) for the corresponding period. The quantity of pollen grains detected outside the MPS, which signifies the difference between the APIn and MPSIn, was considered to detect uncommon pollinations including the autumn pollen integral (AuPIn). These parameters were also studied during the secondary autumn pollen season.

As a complement to the aerobiological data, field phenological data were also used for the present work. The phenological database belonging to our research group included in situ phenological data from the 4 main *Quercus* species growing in the area of Córdoba. The phenological data of flowering were taken periodically during the study years based on the BBCH scale [23]. Phenological surveys were conducted weekly from January to June. During the rest of the year, occasional visits were made, especially in the case of the detection of unusual concentrations of *Quercus* pollen in the aerobiological samplings, as was the case with autumn flowering, when weekly visits were also made [7,24].

### 2.2.2. Meteorological Data

Cordoba has a Mediterranean climate with continental features, low temperatures in winter, and high temperatures in very dry summers, with an annual average temperature of 18.2 °C and a mean annual rainfall of 605 mm, with a predominant wind direction from the southwest (www.aemet.es accessed on 1 January 2020).

Several meteorological variables were analysed, including daily precipitation, expressed in mm (P); rainfall days (Pdays); relative humidity (RH%); hours of sunshine (SunH); daily maximum temperature (Tmax); daily minimum temperature (Tmin); and daily mean temperature (Tmean). Temperature variables were expressed in degrees celsius (°C). Daily meteorological data were obtained from the weather station located at Cordoba airport (37° 50′39″ N, 4° 50′45″ W, 90 m.a.s.l.), AEMet station.

### 2.3. Data Analysis

The potential causes of the autumn *Quercus* pollinations were analysed by performing two consecutive statistical analyses. First, the individual effects of meteorological factors and previous aerobiological variables were tested by applying a Mann–Whitney–Wilcoxon test to identify significant differences among the conditions in the years in which the autumn flowering phenomenon did and did not occur in a univariate manner [25,26].

A multivariate adaptive regression splines (MARS) model with which to explain the autumn pollen integral (AuPIn) was then built. This is a non-parametric regression technique that extends the linear model, incorporating nonlinearities and interactions between variables [27,28]. MARS models do not consider underlying relationships between parameters and can explain linear and nonlinear relationships. The effect of each variable

was quantified using the variable importance parameter, which is a measure of the effect that observed changes to the variable have on the observed response. If two variables are highly correlated, MARS usually drops or underestimates one when building the model. Both variables could have similar importance, but this does not occur with the MARS equation employed to find nonlinear connections between variables in a multivariate manner [27]. The inclusion of different environmental variables and the evaluation of those variables by paying attention to their range of values make it possible to create more realistic models [29]. Of the data, 80 % were used for training and the remaining 20% for external validation, which were randomly selected. The model employed in this study included meteorological variables, pollen season parameters, and variable interactions. The effect of each variable on the autumn flowering events was quantified by estimating the variable importance parameter for each one. This model uses the residual sum-of-squares criterion (RSS) to calculate the effects of variables. This method calculates how the residuals of the model decrease depending on the variables that were omitted; in this way, the importance of each variable is calculated.

The independent variables in the MARS model were the monthly average of the daily mean temperature (°C), the monthly sum of rainfall (mm), the average daily mean temperature during the MPS (°C), the sum of rainfall during the MPS (mm), the starting date of the MPS (day of the year), the finishing date of the MPS (day of the year), the length of the MPS (number of days) and the main pollen season integral (MPSIn) (pollen grains * day/m$^3$).

All the statistical analyses were carried out using the earth package of the R free software, R version 4.2.2, and self-programmed algorithms, which were also employed to construct the models [30,31].

## 3. Results

The characteristics of *Quercus* pollen emissions during the study period were first analysed year by year for the APIn, MPS, and MPSIn. The average MPS in the area was 78 ± SD 18 days (SD: standard deviation), from 17th March ± SD 9 days to 2nd June ± SD 20 days. In addition, the amount of pollen observed from the MPSIn was analysed to determine potential autumn flowering events. An autumn flowering event was considered based on a minimum threshold of 100 pollen grains in Córdoba city cumulated during the autumn period (>100 pollen grains * day/m$^3$ during autumn 22 September to 21 December). Figure 2 shows the absolute range for the highest and lowest historical daily pollen concentrations for a given day, to show readers what the range of possible concentrations was for that day.

Seven years (from the study period 1995–2019) overtook this threshold: 1999, 2002, 2006, 2009, 2012, 2014, and 2015. This threshold was used in order to highlight the real flowering events on the days with pollen records from resuspension. These secondary flowerings occurred from the second half of October until the end of November.

Nevertheless, the intensity of autumn pollination was quite low in comparison with that of spring pollination. It represented an average of 5.12% of the annual pollen integral (APIn), which was 0.66% in the years with less than 100 pollen grains registered during autumn. Regarding the absolute values, an average of 369.1 ± 490.9 pollen grains * day/m$^3$ was detected in the years with autumn flowering vs. 40.5 ± 25.2 pollen grains * day/m$^3$ in the years without. Figure 2 shows both the absolute range of the pollen records during the pollen season and the daily pollen averages in spring (A) and autumn (B).

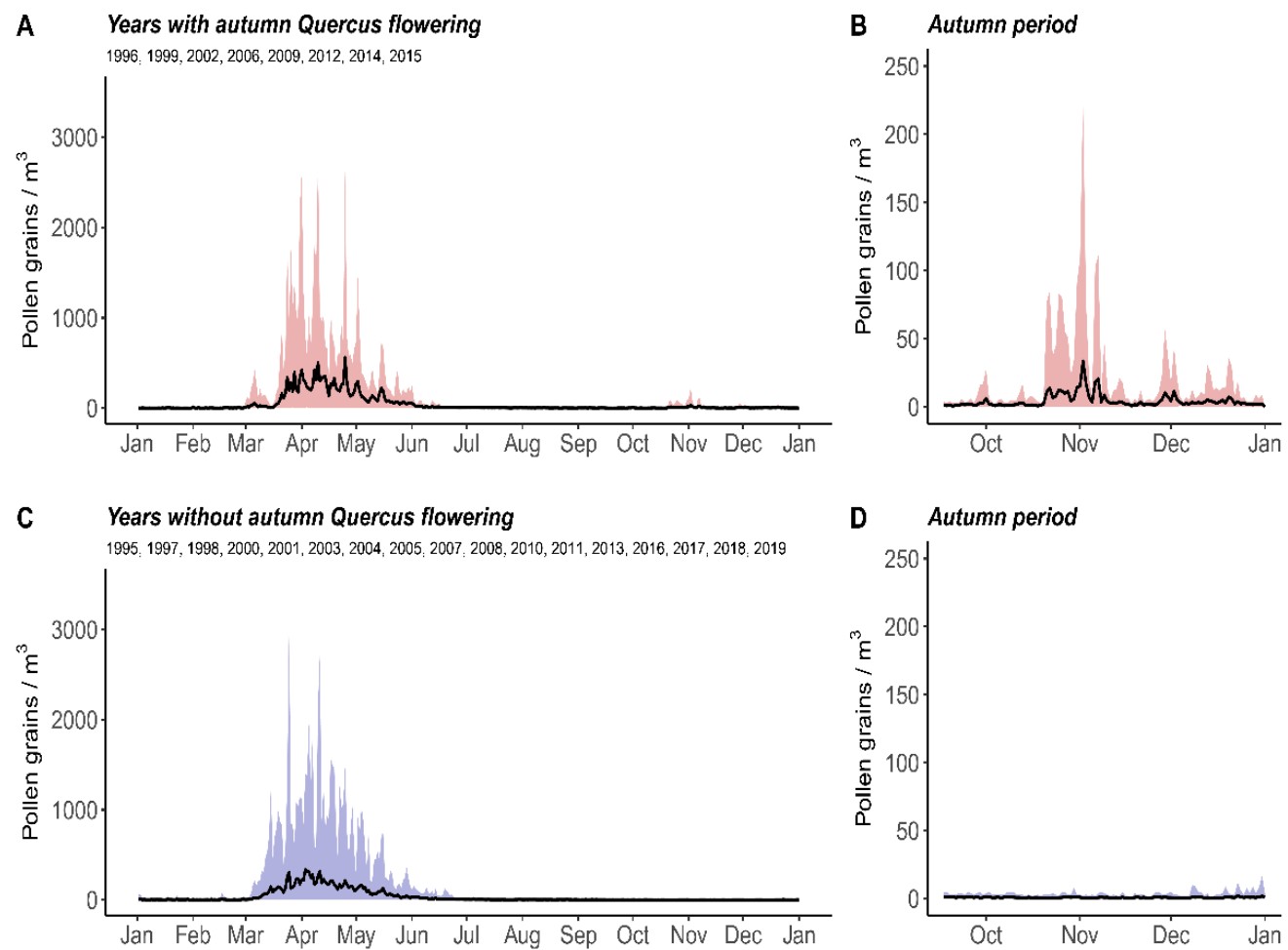

**Figure 2.** Distribution of daily average *Quercus* pollen concentrations in the atmosphere of Córdoba. Black lines represent the average daily pollen concentrations (i.e., average daily pollen concentration on that day of the year during the study period). Shaded areas represent the daily absolute ranges of pollen concentrations (i.e., from absolute minimum "minimum concentration recorded on that day of the year" to absolute maximum "maximum concentration recorded on that day of the year"). Plots A and B refer to the years in which autumn flowering was observed (i.e., autumn pollen integral > 100 pollen grains * day/m$^3$ in Córdoba). Plots C and D refer to the years in which autumn flowering was not observed.

The phenological observations confirmed the autumn flowering of *Q. ilex* subsp. *ballota* in the forests surrounding the city. This flowering did not extend to all individuals. In fact, the percentage of trees that bloomed was an average of 20% of the total. They showed full flowering, including male and female flowers.

Figure 3 shows the results of the Mann–Whitney–Wilcoxon test applied to detect the individual effects of the main meteorological factors on the occurrence of years with and without autumn flowering events. The most remarkable aspect of this figure is the influence of the meteorological conditions in September regarding both the average temperature and the rainfall. A detailed analysis of the climate data during those years indicated that wet and warm conditions in September led to a second flowering, and there was an average threshold of 56 mm of rainfall combined with an average temperature not higher than 25 °C. Warm temperatures during October-November with an average of 17.3 °C were also positive for these second flowering events. Another influencing factor was the conditions in the previous springs. Later spring pollen seasons starting in April and propitiated by dry conditions with around 80 mm of rainfall and high temperatures over 20 °C in the

previous days led to a secondary flowering in autumn. On the contrary, when the MPS started in early or mid-March, this phenomenon did not occur.

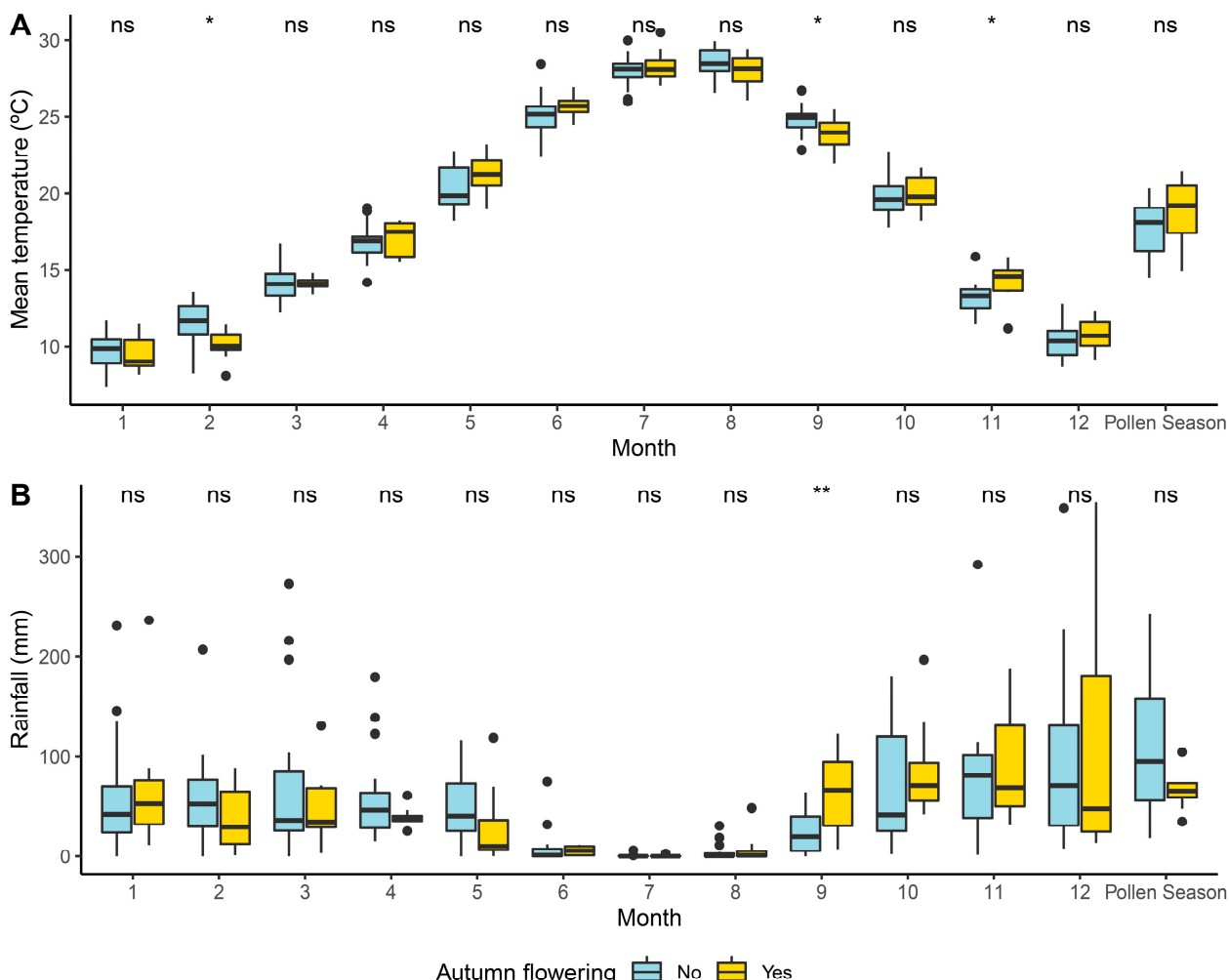

**Figure 3.** Box plot showing the distribution of monthly average (and during pollen season) of daily mean temperature (**A**) and monthly sum of rainfall (**B**) of the years in which an autumn flowering was observed: 1999, 2002, 2006, 2009, 2012, 2014, and 2015. Autumn pollen integral > 100 pollen grains * day/m$^3$, yellow) and years without autumn flowering (blue). Mann–Whitney–Wilcoxon test significance levels: non-significant ($^{ns}$), significant at *p*-value: 0.05 *, 0.01 **.

Figure 4 shows the differences in the features of the previous main pollen season (spring) between the years with and without autumn flowering. Although no significant statistical differences were found, it is possible to observe that those years with a later *Quercus* pollen season in spring (flowering in April is better than in March) presented autumn flowerings in many cases. Fewer pollen grains were also registered during these later pollen seasons in the pre-peak period, i.e., when pollination occurs.

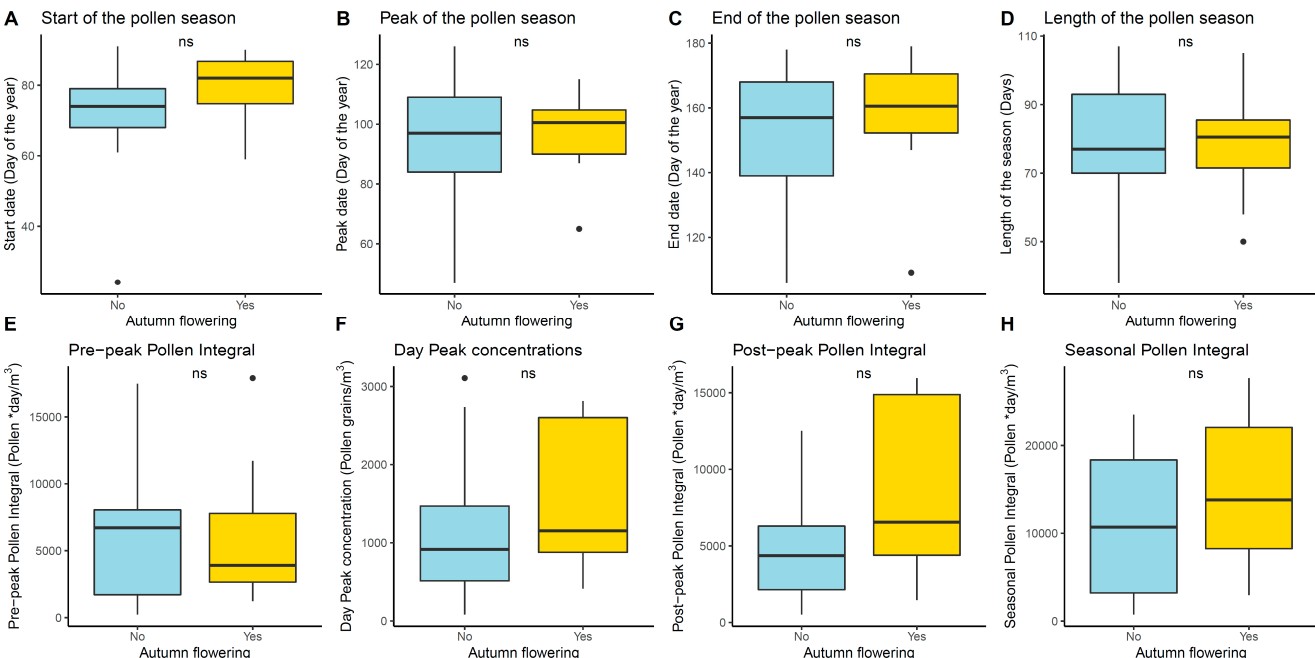

**Figure 4.** Box plot showing the distribution of pollen season features of the main pollen season registered in the springs of the years in which an autumn flowering was observed: 1999, 2002, 2006, 2009, 2012, 2014, and 2015. Autumn pollen integral > 100 pollen grains * day /m³, yellow) and years without autumn flowering (blue). Mann–Whitney–Wilcoxon test significance levels: non-significant ($^{ns}$).

Finally, the multivariate adaptive regression splines (MARS) analysis allowed us to detect the variables controlling autumn flowering. The result was a forecasting model containing three variables: the rainfall in September, the length of the pollen season in the previous spring, and the end of that season, as shown in the following Equation (1).

$$\text{AuPIn} = 27.54 + 36.04 * \max(0, \text{ Rf\_Sep} - 29.5) + 10.29 * \max(0, \text{ end.ps} - 151) - (0.4 * \max(0, \text{ Rf\_Sep} - 29.5) * \text{length.ps}) \tag{1}$$

where AuPIn is the autumn pollen integral (pollen grains * day /m³), Rf_Sep is the accumulated rainfall during September, end.ps is the end of the spring pollen season in Julian days, and length.ps is the length of the spring pollen season in days. The equation has four terms (Figure 5B) and three predictors (Rf_Sep, end.ps, and length.ps). The internal $R^2$ of the validation was 0.93 (Figure 5A). The mean absolute error (MAE) of the model was 59 pollen grains * day/m³ and the external MAE was 128 pollen grains * day/m³. According to the equation, only those years in which the rainfall during September was greater than 29.5 mm were included. Similarly, only those years with a delayed pollen season (ending after 31 May) were considered. However, we observed a significant interaction, that is, the effect of September rainfall, which led to autumn flowerings, exerted less influence if the length of the pollen season was longer than average. The importance of the three significant variables included in the MARS model can be observed in Figure 5C. The variable with the highest weight in the forecasting equation was the September rainfall, with an interaction with the length of the previous spring pollen season. Another relevant variable was the end of the previous spring pollen season. The first two variables had a high degree of forecasting and the combination of both can explain up to 73.6% of the *Quercus* pollen concentration registered in autumn in the Cordoba area (Figure 5D). As can be observed, the years with abundant rainfall during September, together with a late pollen season, were responsible for a relevant flowering event during autumn.

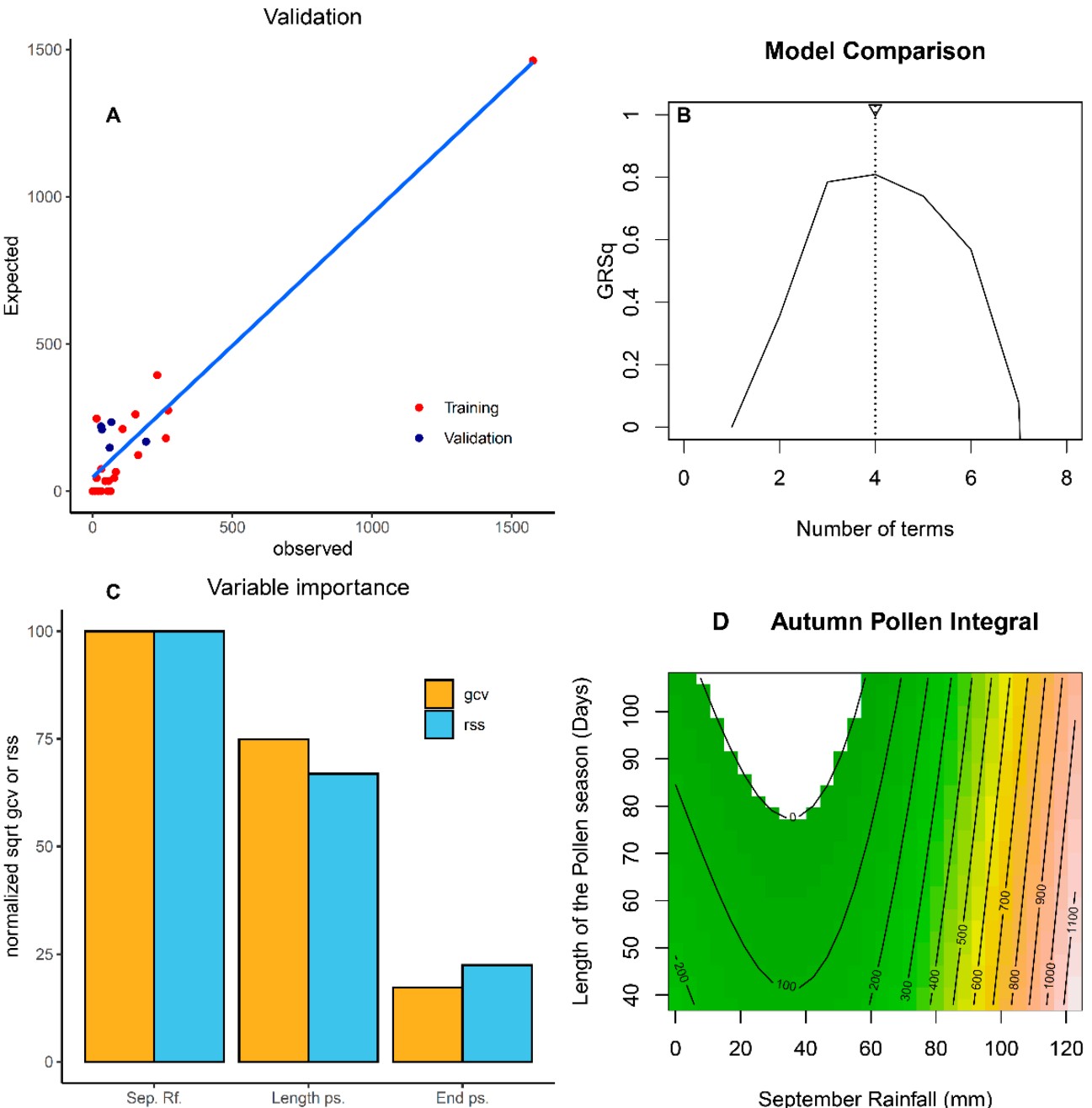

**Figure 5.** Multivariate adaptive regression spline (MARS) model validation of autumn pollen integral (sum of pollen grains * day/m$^3$ during autumn, 22 September to 21 December) (**A**). Selection of number of terms in the model based on GR$^2$ optimization (**B**). Variable importance of impact of GCV (generalized cross-validation) and RSS (residual sum of squares): September rainfall (Sep. Rf.); length of the pollen season (Length ps.), and end of the pollen season (End ps.) (**C**). Combined impact of September rainfall and length of the pollen season on autumn pollen integral for a generalized additive model (GAM) with R-sq: 0.70 and 73.6% of deviance explained with only those two parameters (**D**).

## 4. Discussion

Spring flowering and summer fruiting is the typical reproductive pattern of Mediterranean species, whereas normal autumn flowering is usually related to species with underground storage organs living in continental climates [32]. The occurrence of a second flowering in the corresponding season is an unusual event that has been observed in

some species, where second flowering can occur under certain conditions towards the end of the growing season, which, in the case of the Mediterranean area, coincides with autumn. This leads to extra time for pollination and a second opportunity for successful reproduction, although it is a phenomenon that has been widely ignored in previous research [33]. Until now, there has been no evidence to show whether second flowering events are species-specific or if they are related to specific abiotic or biotic conditions.

In the case of other species, second flowerings have been observed in herbaceous plants overall in mountain environments related to a lack of pollination success in the first flowering, although this hypothesis has not been fully demonstrated [33]. In addition, in the case of shrubs, Castro-Díez and Monserrat-Martí [1] observed a second minimum bloom in Mediterranean forests in summer of *Pistacia* and *Cistus* species, and on some occasions, the flowering was evenly distributed throughout the rest of the year including autumn. In the case of *Cistus,* this phenomenon has been observed by other authors in the Mediterranean area including in southern Spain [34–36]. Castro-Díez and Monserrat-Martí [1] reported that these "out of season" peaks coincided with an increase in temperatures and maximum precipitation in those periods, which are the same conditions that we observed that positively affected *Quercus* autumn flowering during our period of study. Nevertheless, they indicated that occasional flowerings out of season affected less than 20% of the studied populations of *Pistacia* and *Cistus*. These results also coincided with the rate of flowering of the *Quercus* populations observed in our study.

In the case of the *Quercus* genus, sporadic observations of second flowering during late summer and autumn have been reported for *Q. ilex* in coastal areas in southern France (Montpellier) [37] and southern Spain (Málaga) [8]. In the case of the observations published by Recio et al. [8], many of the years they described coincided with autumn flowering in Cordoba. This phenomenon was also observed in other *Quercus* species such as *Q. suber* [8] and *Q. alba* [38]. Nevertheless, as we have observed in the Cordoba area, these studies indicated that in the case of *Quercus* populations blooming twice a year, the greatest bloom occurred in spring and the second one that occurred in autumn was less intense. This fact has been confirmed but the phenological observations revealed that although not all individuals flower, the flowering was complete with male and female flowers. This could be the explanation for the double acorn crop detected in some oak Iberian forestry areas [10].

Nevertheless, the specific factors explaining this phenological behaviour have not been deeply analysed in the reviewed literature. Our analyses revealed that *Quercus* trees were more likely to undergo autumn flowerings in those years with special autumn meteorological conditions, specifically occurring after later spring pollen seasons (April flowering). The most influential meteorological parameters were the September rainfall, followed by warm temperatures during October and November. The conjunction of all these parameters positively influenced the occurrence of a secondary bloom.

Although the Mann–Whitney–Wilcoxon test revealed this influence, the applied multivariate adaptive regression splines (MARS) model was created in order to offer a detailed explanation of the weight of each parameter. In fact, our model explained up to 92% of the autumn pollen integral (AuPIn) based on meteorological variables and previous pollen season parameters. The success of this method was based on the measurable impact of external factors on growth rates in plants, for which each factor usually had a range of optimal values for plant development and also a range of values with a negative impact [39]. MARS models have also been used in other studies in order to relate biological development to environmental conditions and to predict other biological airborne particles such as bacterial endotoxins [40]. They are flexible tools that are used to automate the construction of predictive models by selecting the relevant variables, transforming the predictor variables, processing missing values, and preventing overshooting using a self-test.

Our results indicated that the combined study of previous pollen season features together with meteorological parameters made it possible to understand this atypical ecological behaviour that could be related to a reproductive strategy. In fact, some authors

have revealed that there are trade-offs between reproductive and vegetative growth in *Quercus ilex* individuals and that these occur in annual and intra-annual periods [41]. Our results are consistent with resource-switching and resource-matching hypotheses. Resource switching can occur in different periods and at different ecological levels, depending on the availability of resources such as rainfall and temperature [42,43]. In this sense, in the Mediterranean area, water is a scarce resource and, in the case of dry summers, rainfall in early autumn after a hot summer together with a warm temperature can lead oaks to take advantage of these conditions for flowering. The factors revealed as the most influential would act as the basis for the flowering hormone's function, especially in gibberellins for breaking the dormancy of buds. Our results revealed a capacity to adapt to climatic changes, which have occurred frequently during the history of the Earth, and this could be a relict behaviour that is a consequence of the evolutionary history of Mediterranean species.

## 5. Conclusions

During the 25-year study period, there were 7 years in which a secondary *Quercus* flowering occurred in the province of Cordoba. In our case, the time when these autumn flowerings occurred ranged from the second half of October until the end of November. The univariate statistical analysis of the influence of environmental variables determined that the meteorological conditions during autumn were the most influential. Higher mean temperatures during October-November, together with record rainfall in September, significantly influenced the occurrence of autumn flowering events. The characteristics of the spring pollen season also influenced the occurrence of autumn flowering events. The years with a later *Quercus* pollen season in spring tended to have autumn flowerings. The multivariate statistical analysis made it possible to build a forecasting model based on the combined effects of September rainfall and the properties of the spring pollen season. The degree of explanation was 92%. The validation showed a strong relationship between the expected and the observed autumn pollen concentrations.

The analysis of a long-term *Quercus* pollen database revealed that the main causes of this unusual second flowering in autumn were strongly related to climate change, i.e., strong dry summers and warm autumns. In addition, the results showed that the phenomenon was more frequent in the years with low pollen emissions during spring due to different meteorological events potentiated by climate change such as dryness or heavy rain episodes, as a way of ensuring acorn crops. The results explain how this unusual and lesser-known phenomenon in agroforestry dynamics is related to climate change and the main factors that are driving it, as well as the potential consequences for these important and endangered Mediterranean ecosystems.

**Author Contributions:** Conceptualization, H.G.-M.; methodology, J.O.; software, J.O.; validation, J.O.; investigation, R.L.-O.; analysis of the data, R.L.-O. and J.O.; writing—original draft preparation, H.G.-M.; writing—review and editing, R.L.-O., J.O., and C.G.; supervision, C.G.; project administration, H.G.-M.; funding acquisition, H.G.-M. All authors have read and agreed to the published version of the manuscript.

**Funding:** This research was funded by the CLIMAQUER project (Reference 1260464), awarded by the Ministry of Economy and Knowledge of the Andalusian Regional Government through the European Regional Development Funds (ERDF).

**Data Availability Statement:** The datasets presented in this article are not readily available since the data are private. Requests to access the datasets should be directed to Carmen Galán Soldevilla at bv1gasoc@uco.es.

**Acknowledgments:** The authors wish to thank the CLIMAQUER project (Reference 1260464) awarded by the Ministry of Economy and Knowledge of the Andalusian Regional Government through the European Regional Development Funds (ERDF) for its support.

**Conflicts of Interest:** The authors declare no conflict of interest.

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
