# Peer review of "Factors Driving Autumn Quercus Flowering in a Thermo-Mediterranean Area"

_agronomy, doi:10.3390/agronomy12112596_

Round 1

Reviewer 1 Report

The authors explored the parameters that could explain a secondary flowering period of Quercus species in Spain. They based their analyses on an impressive 25-year air pollen content daily record data set. They provide a predictive model with a degree of explanation of 92%. This analysis reveals that the autumn second Quercus flowering is strongly linked to climate change (strong dry summer and warm autumns).

This study provides strong evidence to predict incoming phenological dis-regulation linked to climate change.

The text and figures are clear and comprehensive. The conclusion provides clear take-home messages.

Even if the authors evoked the question with a reference, the study would have benefited from phenological field observations as numerous questions remain unsolved. For example, does the autumn pollen production remain low because only a few trees bloom with a lot of flowers, or many trees with very few flowers? Are they also female flowers? Does the autumn flowering event impact the level of subsequent spring flowering? Is the autumn pollen of the same quality as spring pollen? Such questions could be evoked in the discussion section and developed in another paper!

I have very few specific comments:

L. 36 "us" should it be "as"; There is a double space before "Results."

L. 66 "pollination events", you can not say that. You only reported here pollen production events. We have no clues about any pollination (are female flowers? Is viable pollen effectively arrive on receptive females flowers of the same species, etc.)

L. 81 "are already unknown" maybe need to be rephrased 

L. 84 You said "four specific objectives", but you presented only three…

L. 349 Please remove the first "and".

Fig 3 and 4, please specify the years in each category.

Author Response

We explain below all the corrections we made indicating all the changes and revisions made following the reviewers’ suggestions. Reviewers’ comments are in italics. Line numbers are here detailed, and in the Manuscript, changes are pointed out in blue.

Corrections made following the reviewers’ comments.

Reviewer 1

The authors explored the parameters that could explain a secondary flowering period of Quercus species in Spain. They based their analyses on an impressive 25-year air pollen content daily record data set. They provide a predictive model with a degree of explanation of 92%. This analysis reveals that the autumn second Quercus flowering is strongly linked to climate change (strong dry summer and warm autumns).

This study provides strong evidence to predict incoming phenological dis-regulation linked to climate change.

The text and figures are clear and comprehensive. The conclusion provides clear take-home messages.

Even if the authors evoked the question with a reference, the study would have benefited from phenological field observations as numerous questions remain unsolved. For example, does the autumn pollen production remain low because only a few trees bloom with a lot of flowers, or many trees with very few flowers? Are they also female flowers? Does the autumn flowering event impact the level of subsequent spring flowering? Is the autumn pollen of the same quality as spring pollen? Such questions could be evoked in the discussion section and developed in another paper!

                     As we indicate in the original paper, field phenological sampling was made during all the study time. Nevertheless, more information about is now given both in the Material and Methods (Lines 145-152) and in the Results (230-233) sections.  This fact is also discussed in the Dicussion section (Lines 335-338). There, the questions of the referee are answered.      

Specific comments:

  1. 36 "us" should it be "as"; There is a double space before "Results."

Made

  1. 66 "pollination events", you can not say that. You only reported here pollen production events. We have no clues about any pollination (are female flowers? Is viable pollen effectively arrive on receptive females flowers of the same species, etc.)

We have changed pollination for pollen season or flowering through all the manuscript.

  1. 81 "are already unknown" maybe need to be rephrased 

We have rewritten that phrase. Now is better understandable.

  1. 84 You said "four specific objectives", but you presented only three…

Corrected. Now 3 are indicated.

  1. 349 Please remove the first "and".

Made

Fig 3 and 4, please specify the years in each category

Made

Reviewer 2 Report

I am not familiar with the flowering procedures described in the current study, but the general statement appears to be appropriate and useful. In recent years, researchers put a lot of effort into extreme climate events and their thorough impacts. Hence, insightful analysis of climate effects on flowering is valuable to determine possible routes. The sheer amount of work (including 25 years of long-term pollen monitoring) that went into doing the analyses for samples, and on this count alone deserves publication. However, I think there are some rough indications for this paper to be published. I recommend a major revision.

 I have the following comments to improve clarity.

Please see comments for specifics.

Figure1 Legend color should be unified (green color) and a bit ambiguous in the current version. A higher resolution figure should be provided as well.

2.2.2 Meteorological data Line158, the location mentioned should be listed in figure1 as well.

2.3 Data analysis Line 163, citations or more indications should be stated to indicate the solidity of the current analysis

Line 178 “We have already … of olive pollen” please reword the sentence.

Line 185 a simple example would be helpful to indicate the RSS approach

Figure 2 The reason to put out the absolute range of pollen concentration should be indicated precisely. 

The discussion section needs major revisions. Much of the discussion is a literature review rather than an expansion of the paper’s findings and how they relate to pass research or how they could be applied.

 I hope these comments are useful.

Author Response

We explain below all the corrections we made indicating all the changes and revisions made following the reviewers’ suggestions. Reviewers’ comments are in italics. Line numbers are here detailed, and in the Manuscript, changes are pointed out in blue.

Reviewer 2

I am not familiar with the flowering procedures described in the current study, but the general statement appears to be appropriate and useful. In recent years, researchers put a lot of effort into extreme climate events and their thorough impacts. Hence, insightful analysis of climate effects on flowering is valuable to determine possible routes. The sheer amount of work (including 25 years of long-term pollen monitoring) that went into doing the analyses for samples, and on this count alone deserves publication. However, I think there are some rough indications for this paper to be published. I recommend a major revision.

I have the following comments to improve clarity.

Figure1 Legend color should be unified (green color) and a bit ambiguous in the current version. A higher resolution figure should be provided as well.

The legend color change depending on the Quercus species. Better resolution is now given to the images.

 2.2.2 Meteorological data Line158, the location mentioned should be listed in figure1 as well.

Made. Coordinates are now given.

 2.3 Data analysis Line 163, citations or more indications should be stated to indicate the solidity of the current analysis.

Now references about the strength of the Wilconxon-Mann-Withney analysis are given Line 169.

 Line 178 “We have already … of olive pollen” please reword the sentence.

That phrase was not so appropriated in that section. We have rephrased it. Lines 180-182.

Line 185 a simple example would be helpful to indicate the RSS approach

Now the RSS criterion has been better explained for a better understanding. This method is used to calculate the weight of each variable by measuring the error if that variable is not included. Lines 187-189.

Figure 2 The reason to put out the absolute range of pollen concentration should be indicated precisely. 

The absolute range displays the highest and lowest historical daily pollen concentration for a given day, to show readers what the range of possible concentrations is for that day. It is explained in Lines 206-209 and in the Figure 2 legend.

The discussion section needs major revisions. Much of the discussion is a literature review rather than an expansion of the paper’s findings and how they relate to pass research or how they could be applied.

The discussion section has been fully revised. Now our results are deeply discussed and less “introduction” appear. Main changes are pointed out in the text. They are mainly in Lines 309-326, Lines 335-349 and Lines 359-374. These paragraphs have been revised according with the reviewers’ suggestions.

In this way conclusions have been revised, Lines 389-397.

Reviewer 3 Report

Dear Authors,

comments and recomandations are inclosed in the pdf file revised.

Author Response

We explain below all the corrections we made indicating all the changes and revisions made following the reviewers’ suggestions. Reviewers’ comments are in italics. Line numbers are here detailed, and in the Manuscript, changes are pointed out in blue.

Reviewer 3.

This reviewer gave us the revisions on a pdf document. We detail the suggestions and changes as follows:

Line 52. An average period must be indicated.

Made.

The paragraph about the time of monitoring and the place should be moved.

Made to Lines 78-84.

Lines 119. In the M&M section the location of Quercus populations regarding the city should be indicated.

Made. Also. the Figure 1 is now with a better resolution.

How the Quercus pollen grains were identified? The traps could contain other pollen species. A summary explanation is needed.

In Lines 123-125 is explained that pollen grains were identified under a light microscopy following the rules of the European Aerobiology Society rules. In our laboratory we count all the pollen grains in the aerobiological samplings taken by the traps. In this case we analyse Quercus pollen data.

In my opinion the term 'pollination' is incorrect. Indeed, the pollination process was not studied. It could be better define it as 'pollen emission or presence'. this observation is valid throuhgout the text

This expression has been changed by “pollen emission” through all the manuscript.

The phenological stage have to be added, i.e. BBCH scale.

The phenological scale used was the international BBCH one. The reference has been included (Line 149). Moreover more explanations about the phenological survey are given in Lines 145-152.

Lines 235-247. This paragrph needs to be clarified with major explanations about the relationship between temperatures and rainfall are needed. For example, what about the significance in february in fig.A?

More explanations about the significance of the different variables are now given.

Figure 3 must include the term “Pollen Season” in the X axis.

Made

Lines 255-260.  The paragraph was so speculative.

We rewrite that paragraph to avoid speculations and for a better understanding.

In Figure 4 add titles at the top of graphs

Made.

Lines 283-285, sentence not clear, rewrite

Made.

Too long introduction. In the discussion section, rewrite.

The discussion section has been fully revised. Now our results are deeply discussed and less “introduction” appear. Main changes are pointed out in the text. They are mainly in Lines 309-326, Lines 335-349 and Lines 359-374. These paragraphs have been revised according with the reviewers’ suggestions.

What about the later period? Add a range.

Made. Lines 343-345.

Line 347. How this phenomenon could be explained from a physiological point of view? A balance between hormonal and nutrient compounds? Please, discuss this aspect.

All this paragraph has been rewritten Lines 360-375. Now a deep discussion from a physiological point of view is given, including hormones and dormancy aspects.

Last paragraph of the conclusions’ section is not clear.

In this way conclusions have been revised, Lines 389-397.

Round 2

Reviewer 1 Report

The authors nicely reworked their paper, adding all the requested precisions and modifications. The study is now very pleasant to read, and the data set of 25 years is still as impressive.

I jute have a few tiny remarks.

• Aren't pollen integrals should be expressed as pollen grains * day/m3 instead of pollen * day/m3?

L 149 correct the double space before "Phenological"

L 182 "80%" that starts the sentence should be in letters

L 206 correct m3 to m3

L210 "7" that starts the sentence should be in letters Seven

L 241 correct "56.mm"

L 348 correct the double space after "applied"

Author Response

DEPARTMENT

Botany, Ecology and Plant Physiology

 Carmen Galán Soldevilla
Dpt. of Botany, Ecology and Plant Physiology,

University of Cordoba
14071, Cordoba, Spain

October 15th, 2022

Dear Editor,

We wish to submit the revised version (R2) of the original research article entitledFactors driving autumn Quercus flowering in a thermo-Mediterranean area for consideration by Agronomy journal. The manuscript was previously submitted to the journal, revised by 3 reviewers and the decision was Minor Revisions and the editor encouraged us to re-submit it after revision. I confirm that this work is original and has not been published elsewhere, nor is it currently under consideration for publication elsewhere. We explain below all the corrections we made following the reviewers’ comments.

Corrections made following the reviewers’ comments.

Reviewer 1

The authors nicely reworked their paper, adding all the requested precisions and modifications. The study is now very pleasant to read, and the data set of 25 years is still as impressive.

I jute have a few tiny remarks.

  • Aren't pollen integrals should be expressed as pollen grains * day/m3 instead of pollenday/m3?

Yes, now all these expressions have been revised and expressed as pollen grains* day/m3 

L 149 correct the double space before "Phenological"

Revised

L 182 "80%" that starts the sentence should be in letters

Now it is expressed as: The 80% of the data….

L 206 correct m3 to m3

Corrected.

L210 "7" that starts the sentence should be in letters Seven

Made

L 241 correct "56.mm"

Corrected

L 348 correct the double space after "applied"

Corrected

Reviewer 2
Dear Authors,

thank you for accepting the suggestions througout the manuscript. Only a note about the text at the end of the cover letter, it is another journal.....

Ok, apologize for the mistake. It has been corrected.

            ---------------------------------------------

We believe that after these corrections following the reviewers’ suggestions, now the manuscript can be accepted in the Agronomy journal. We have no conflicts of interest to disclose.

Please address all correspondence concerning this manuscript to [email protected].

Thank you for your consideration of this manuscript.

Sincerely,

                        Carmen Galán Soldevilla

Reviewer 3 Report

Dear Authors,

thank you for accepting the suggestions througout the manuscript. Only a note about the text at the end of the cover letter, it is another journal......

Author Response

(The authors gave the same response as above.)
